biomechanics

locomotion, vision, biomechanics, kinematics, obstacle traversal

**Author for correspondence:**
Katherine A. J. Daniels
e-mail: k.daniels@mmu.ac.uk

# Visuomotor control of leaping over a raised obstacle is sensitive to small baseline displacements

Katherine A. J. Daniels[1,2] and J. F. Burn[1]

[1]Queen's School of Engineering, University of Bristol, Bristol, UK
[2]Department of Sport and Exercise Sciences, Musculoskeletal Science and Sports Medicine Research Centre, Manchester Metropolitan University, Manchester M15 6BH, UK

KAJD, 0000-0001-8134-6764

The limb kinematics used for stepping or leaping over an obstacle are determined primarily by visual sensing of obstacle position and geometry. In this study, we demonstrate that changes are induced in limb kinematics even when obstacle geometry is manipulated in a way that does not introduce a mechanical requirement for a change of limb trajectory nor increase risk of collision. Human participants performed a running leap over a single raised obstacle bar. Kinematic changes were measured when an identical second bar was introduced at a ground level underneath the obstacle and displaced by a functionally insignificant distance along the axis of travel. The presence or absence of a baseline directly beneath the highest extremity had no significant effect on limb kinematics. However, displacing the baseline horizontally induced a horizontal translation of limb trajectory in the direction of the displacement. These results show that systematic changes to limb trajectories can occur in the absence of a change in sensed mechanical constraints or optimization. The nature of visuomotor control of human leaping may involve a continuous mapping of sensory input to kinematic output rather than one responsive only to information perceived to be mechanically relevant.

## 1. Introduction

Jumping and leaping from a running gait are manoeuvres in the locomotor repertoire of many species of terrestrial animals. These manoeuvres increase the versatility of locomotion by allowing raised obstacles in the path of travel to be traversed without re-routing or large changes to speed of travel. The control of limb trajectories to avoid contact with raised obstacles requires anticipatory control using sensory input from vision [1].

Running humans adjust their stride length 4–6 steps in advance of the obstacle to place the foot in a suitable position for take-off [2–4]. The body is then lowered and leg stiffness modulated in the last two steps before take-off to facilitate the vertical redirection of the body centre of mass [5,6]. Finally, both anticipatory and online control of lower limb trajectories are used during the flight phase to avoid any part of the body colliding with the obstacle [7,8].

The way in which the visual scene is sampled and interpreted to control jumping and leaping remains unclear but is of interest to biologists, designers of the built environment and engineers developing leg-based vehicles. Studies of eye movements during a variety of tasks have found that fixations are consistently directed to task-relevant regions of the environment to provide visual information for control of future action [9–16]. From a mechanical perspective, a relatively small amount of spatially localized information defining the geometry and location of an obstacle is necessary and sufficient to plan a leap trajectory over it [17], and a plethora of experimental studies have demonstrated the close relationship between obstacle geometry and traversal kinematics [7,8,18,19]. In obstacle traversal tasks, it has been found that saccades are used during the approach to selectively target spatial regions from which this information might be obtained [15,20,21].

Visual information from the wider field of view, while not strictly necessary to avoid contact with the obstacle, has been found to modulate limb kinematics. When walking over raised obstacles, foot trajectories have been observed to change if the visual properties of the obstacle and surrounding environment are altered to influence the perceived consequence of a fall [22] or fragility of the obstacle [23]. Limb kinematics have also been found to be altered by the visual properties of material placed vertically below the top edge of a raised obstacle [24–29] or to the sides of the top edge [29–31], although control apparently modulated only by the position of the top edge has been observed in visually rich environments [32]. In these studies, manipulations were made either to information hypothesized to alter risk perception, or to information spatially close to a geometric feature that is necessary to determine height. The resulting changes to limb trajectories are interpreted as gait adaptations intended to facilitate successful traversal after perceived alteration to obstacle height or to the destabilizing consequences of accidental foot–obstacle contact. However, not all geometric manipulations to a raised obstacle would be expected to alter these percepts. When walking over a vertical obstacle the closest foot placements on both the take-off and the landing side are 20–30 cm distant from the base of the obstacle [23,29,33], and these distances can exceed 1 m when leaping over an obstacle from a running gait [34]. Small manipulations to geometry at the base of the obstacle do not, therefore, necessitate modifications to foot placements or traversal limb trajectories, particularly when traversing the obstacle from a faster gait. It is unknown whether gait is adapted in response to such manipulations, or whether locomotor control mechanisms for obstacle traversal are responsive only to information that alters the sensed mechanical constraints.

The aim of this study was to investigate whether a simple geometric manipulation, spatially remote from the geometric feature signalling height and not affecting the perception of risk, would still elicit a change in limb kinematics during a running leap manoeuvre. Firstly, we presented human participants with a single pole raised from the ground to confirm that sufficient information could be obtained from it to execute a leap. A safe trajectory required knowledge only of the location of the pole in the horizontal plane and its height above ground [17], both of which could be obtained from the pole using visual cues such as convergence, accommodation, optical flow, motion parallax and binocular disparity during the approach [35]. Secondly, we introduced a second identical pole on the ground in three positions which, from a mechanical perspective, would not require any change to the traversal kinematics used for the single pole. As no change in trajectory was required, or indicated by a consideration of safety margin, an observed change would imply not only that the control of the manoeuvre incorporated information about the second pole, but also that these changes could not be reasonably explained by consideration of mechanical constraints and risk.

# 2. Materials and methods

## 2.1. Participants

Fourteen participants were recruited for the study (12 female and two male, age range 19–31 years, body mass  mean ± s.d.  68.8 ± 14.7 kg,  standing  greater  trochanter  height  mean ± s.d.  0.83 ± 0.04 m). Participants' level of participation in sporting activities varied but none competed at or above university level in any athletic jumping event. All were free from self-reported musculoskeletal

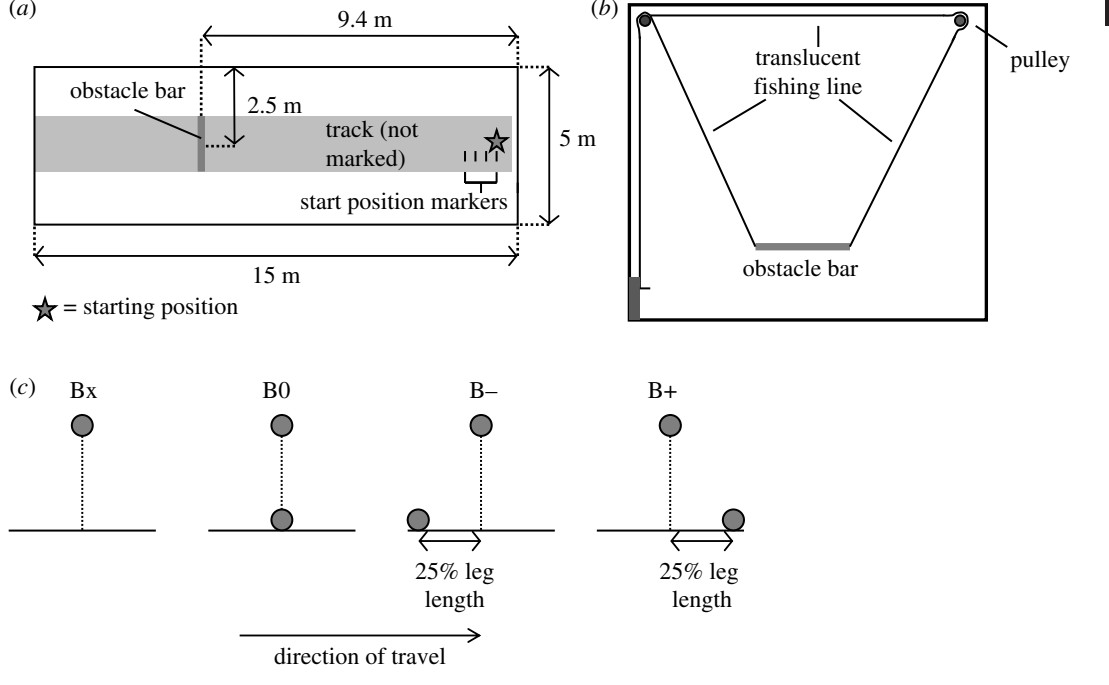

**Figure 1.** Experimental set-up (not to scale). (*a*) Plan view of laboratory showing the position of obstacle and trial starting position of participants. (*b*) Front view looking along the track from the starting position showing obstacle suspension mechanism. (*c*) Obstacle baseline conditions: lateral view of obstacle.

injuries/impairments and visual impairments. The protocol was approved by the University of Bristol faculty ethics committee and all participants gave informed written consent.

Fourteen retro-reflective spherical markers of 14 mm diameter were attached to anatomical landmarks to define an 11-segment (head and trunk, upper arms, forearms, thighs, shanks, feet) rigid body model [36]. Additional markers were placed on the trochlear process of the calcaneus (*Heel*) and distal interphalangeal joint of the second phalanx (*Toe*). The distance along the long axis of each foot from the Heel marker to the posterior extremity of the heel (*offset*$_{Heel}$) and from the Toe marker to the distal extremity of the phalanges (*offset*$_{Toe}$) were measured during standing. The offset measurements were used to calculate distances between the nearest extremity of the foot and the baseline during stance phases but were not used to generate virtual markers in the swing phase to avoid making assumptions about the orientation of the distal phalanges. All participants performed the experiment barefoot.

## 2.2. Data collection

The experiment took place in a 15 m long × 5 m wide indoor gait laboratory with a flat, level floor. The flooring surface was a non-slip, non-reflective, grey-coloured proprietary material with no visible texture or features. A central region 1 m wide spanning the full length of the laboratory and located equidistant from the side walls was defined as the *track* for the purpose of description but was not physically delimited. Four start location markers were placed on the track at the end from which data capture trials commenced, spaced 0.3 m apart in the direction of travel (figure 1*a*).

An *obstacle bar* was constructed by suspending a 1 m length of reinforced polystyrene piping over the track using a thin translucent fishing line (figure 1*a,b*). The suspension apparatus was not visible when approaching or traversing over the obstacle. The height of the top of the bar, defined as *obstacle height*, was set to 0.55 × participant leg length (low obstacle) and 0.60 × participant leg length (high obstacle). These resulted in obstacle heights of 0.41–0.50 m (low) and 0.45–0.54 m (high). Obstacle height as a proportion of leg length, a dimensionless number, was termed *relative obstacle height* (ROH). Two different heights were used, and presented in a random sequence during data collection, to minimize the likelihood of a learning effect.

A bar identical to that suspended was positioned underneath the obstacle to form a baseline. The experimental conditions were created by offsetting the baseline from the obstacle bar. Four baseline positions were used (figure 1c):

*Condition 1* (*Bx*): No obstacle baseline was present in the proximity of the obstacle bar, but the baseline bar was placed at the end of the track against the wall. This was in order that the same physical sources of visual information were available for all trials, although, in this condition, the baseline bar was not part of or relevant to the obstacle.
*Condition 2* (*B0*): Baseline positioned vertically below the obstacle bar (zero offset).
*Condition 3* (*B+*): Baseline positioned 0.25 leg length offset on the landing side of the obstacle bar.
*Condition 4* (*B−*): Baseline positioned 0.25 leg length offset on the approach side of the obstacle bar.

An offset of 0.25 leg length was chosen because it was found during preliminary work to result in baseline positions that would not interfere mechanically with the normal foot placements before and after an obstacle when an obstacle with a zero-offset baseline (B0) was traversed from the target approach speed.

Participants practised running on the track with no obstacle present until they were able to maintain a speed of approximately Froude number (Fr) 1, where $Fr = v/(gl)^{0.5}$ and $v$ represents speed (m s$^{-1}$), $g$ represents acceleration due to gravity (m s$^{-2}$) and $l$ represents leg length (m). They were instructed to run at this speed (equivalent to 2.7–3.0 m s$^{-1}$) when approaching the obstacle. Participants were randomly allocated a start position for each trial to minimize the likelihood of a systematic relationship between obstacle position and the unmodified stride cycle spatial kinematics at the time of encounter. The toe of the named foot was placed level with the named start position marker and the contralateral leg protracted to initiate the first step of the trial.

All obstacle traversal trials began with the participant at the approach end of the track with their back to the obstacle, so as not to see obstacle manipulations made between trials. The participant turned around when verbally cued by the experimenter, positioned their foot at the relevant start position marker, ran along the track at the target speed, leapt over the obstacle and continued running to the far end of the track. Motion capture data were collected at 120 fps. Two B0 practice trials were permitted prior to data collection, one at ROH 0.55 and one at ROH 0.60. The participant then performed 32 trials comprising 4 trials of each obstacle height–condition combination presented in a random order.

## 2.3. Analysis

### 2.3.1. Calculation of kinematic parameters

Kinematic data were filtered using a fourth-order zero phase shift Butterworth low-pass filter with a corner frequency of 9 Hz. This filter configuration is commonly used for smoothing human gait kinematic data and has been applied in studies of long jump kinematics [37–40].

Stance phases were identified automatically using an algorithm incorporating vertical height and horizontal velocity of a marker placed on the fifth metatarsal head of the relevant foot (horizontal velocity of marker less than 0.006 m s$^{-1}$ and vertical height less than 0.05 m). Foot contact with the ground was used to define the beginning of a step. The step containing the ultimate stance phase on the take-off side of the obstacle was defined as step $n$ with previous steps termed step $n-1$, $n-2$, … and following steps termed $n+1$, $n+2$, …. The *lead leg* was defined as the first foot to contact the ground on the landing side of the obstacle and the *trail leg* as the second foot to contact the ground on the landing side of the obstacle.

Parameters used for analysis describe the key features of the trajectory of the lead and trail leg trajectories over the obstacle and are illustrated in figure 2 with definitions in table 1. The Toe marker was used to define foot position for all parameters except obstacle clearance. For obstacle clearance, the vertical distance from the top of the obstacle to both the Toe and the Heel markers were recorded and the minimum of these values was reported. The associated marker was termed the *clearance marker*.

All kinematic parameters used for analysis were converted to dimensionless form by dividing those with the dimensions of length by leg length, $l$, and those with the dimensions of speed by $(gl)^{0.5}$.

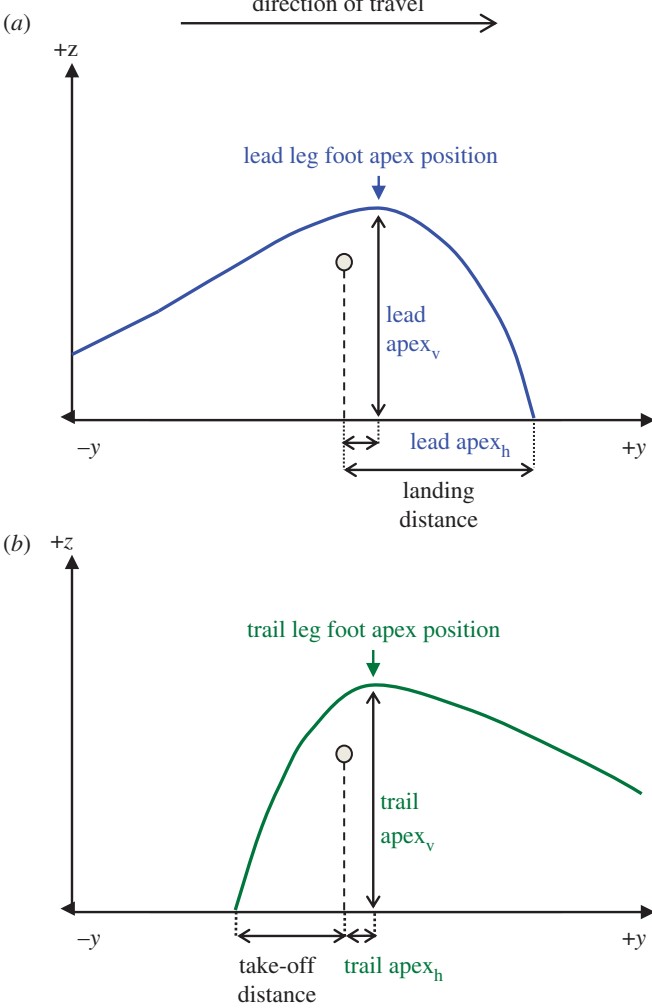

**Figure 2.** Schematic lateral view diagram of lead leg (top) and trail leg (bottom) foot trajectories and the obstacle traversal parameters measured. Trajectory apex positions are described in the $y$ (horizontal; direction of travel) axis and $z$ (vertical) axis, where ($y = 0$, $z = 0$) is located at ground level vertically below the top edge of the obstacle. Further definitions and calculations are given in table 1.

### 2.3.2. Statistical analysis

It was first verified that the displaced baseline was unlikely to have induced a mechanical requirement to alter foot trajectories during obstacle traversal by comparing the step $n$ and $n+1$ stance foot placement restrictions enforced in conditions B− and B+ with the take-off and landing foot placements actually recorded during B0 trials. Stance foot offset$_{Toe}$ and offset$_{Heel}$ were added or subtracted from the relevant marker positions to calculate the distance from the nearest extent of the foot to the baseline in each case.

Of primary interest was whether baseline condition affected traversal kinematics. The effect of obstacle height and trial number were also investigated to identify potential interactions. Three-factor within-subjects repeated-measures ANOVA with obstacle height (low, high), baseline condition (Bx, B0, B−, B+) and trial number (1–4) as factors were used for each dependent variable to test the null hypotheses that these factors had no effects on foot trajectories during obstacle traversal. Greenhouse–Geisser's adjustment to the degrees of freedom was used when Mauchly's test of sphericity was significant, and Tukey's HSD *post hoc* tests were used to identify pairwise differences between groups when main effects were found. Bartlett's test for equality of variances was used to test for the equal variance of the dependent variable between condition Bx and B0 for each obstacle height. Significance was accepted at $\alpha = 0.05$ and the symbols *, ** and *** were used in figures to represent $p < 0.05$, $p < 0.01$ and $p < 0.001$, respectively. No family-wise error rate correction was performed for ANOVA main effects because these analyses were planned and complementary.

**Table 1.** Parameters used for analysis. Key features of the traversal and of the lead and trail foot trajectories. All position and distance parameters are reported relative to leg length; speed parameters are reported as Froude numbers.

| parameter | definition |
| --- | --- |
| lead clearance, trail clearance | smallest vertical distance between the raised obstacle bar and the foot during traversal steps of the lead and trail legs, respectively, when $y = 0$ |
| step length | the total length of the obstacle traversal step equal to the sum of take-off distance and landing distance |
| landing distance | horizontal distance from the raised obstacle bar to the Toe marker of the lead foot during step $n+1$ stance phase |
| take-off distance | horizontal distance from the Toe marker of the trail foot to the raised obstacle bar during step $n$ stance phase |
| lead apex$_h$, trail apex$_h$ | horizontal distance from the raised obstacle bar position ($y = 0$) to the apex of the foot trajectory, for lead and trail legs, respectively |
| lead apex$_v$, trail apex$_v$ | vertical distance from $z = 0$ to the apex of the foot trajectory, for lead and trail legs, respectively |
| traversal speed | traversal step speed, calculated as $v/\sqrt{gl}$ where $v = (\text{step } n \text{ length})/(\text{step } n \text{ time})$ |
| approach speed | approach step speed, calculated as $v/\sqrt{gl}$ where $v = (\text{step } n-2 \text{ length})/(\text{step } n-2 \text{ length})$ |

Finally, take-off distance and step length were plotted against approach speed and traversal speed to visualize the relationship between approach/traversal speed and the positions of the take-off and landing foot placements. Linear regression models were fitted to the data to determine the proportion of linear variation in take-off distance and step length that was due to variation in speed of travel.

## 3. Results

A total of 448 trials were collected from 14 participants (112 at each obstacle height). No obstacle contacts occurred. Motion capture data were recorded for steps $n-2$, $n-1$, $n$ and $n+1$ of all trials.

For all B0 trials, the mean (±s.d.) take-off distances were $0.72 \pm 0.02$ (high obstacle) and $0.67 \pm 0.02$ (low obstacle). The landing distances were $0.50 \pm 0.02$ (high obstacle) and $0.52 \pm 0.02$ (low obstacle). All distances were significantly greater than the baseline displacement distance (0.25) in B− and B+ and so we were able to conclude that baseline displacement imposed no mechanical requirement for a systematic change in traversal kinematics.

Parameters were grouped into those describing the horizontal ($y$-axis) and those describing the vertical ($z$-axis) positions of the measured foot trajectory features (figure 2). There were no significant interactions between any combination of baseline condition, height and trial number and no significant effect of trial number for any measured parameter (all $p > 0.05$). The main effects are thus reported below.

### 3.1. Effect of baseline condition on horizontal distances

No main effect of baseline condition on step length was observed ($F(3, 39) = 2.24$, $p = 0.098$; figure 3*a*).

There were main effects of baseline condition on both take-off and landing distances ($F(3, 39) = 4.67$, $p = 0.007$ and $F(3, 39) = 14.14$, $p < 0.001$, respectively). Take-off distance decreased (figure 3*b*) and landing distance increased (figure 3*c*) as the baseline moved from the take-off to the landing side of the obstacle (B− → B0 → B+). No significant differences in the mean or variance of take-off or landing distances were found between Bx, the condition in which no baseline was present, and B0, baseline present with zero offset.

The lead leg apex was generally on the landing side of the obstacle (greater than 0) and was affected by baseline condition ($F(3, 39) = 6.65$, $p = 0.001$; figure 3*d*), shifting in the same direction as the baseline was moved. Trail leg apex was generally on the take-off side of the obstacle (less than 0; figure 3*e*) and

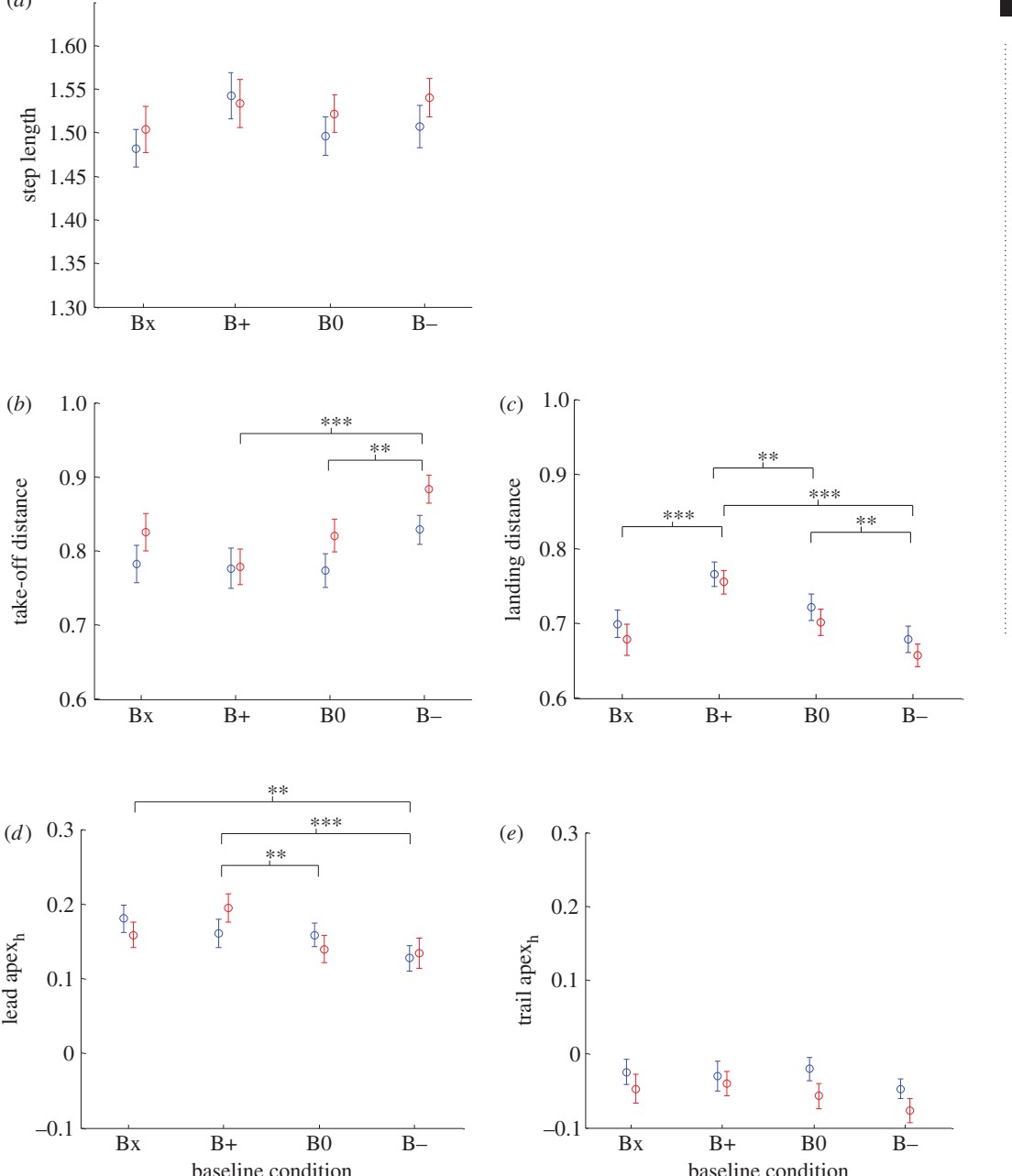

**Figure 3.** Effect of obstacle condition on horizontal parameters. (*a*) Step length for the four obstacle conditions. Take-off distance (*b*) and landing distance (*c*) for the four obstacle conditions. Lead (*d*) and trail (*e*) foot peak apex horizontal positions for the four obstacle conditions. Circles indicate the mean value of the measured parameter for low obstacle crossings (blue) and the mean value of the measured parameter for high obstacle crossings (red). Error bars show ± s.e.m. The symbol $^*$ indicates a significant difference between baseline conditions at $p < 0.05$, $^{**}$ at $p < 0.01$ and $^{***}$ at $p < 0.001$.

was not significantly affected by obstacle position ($F(3, 39) = 2.83$, $p = 0.053$). No significant differences in the mean or variance of lead apex$_h$ or trail apex$_h$ were found between Bx and B0.

Only take-off distance was found to be affected by obstacle height and was greater for high obstacles than for low obstacles ($F(1, 13) = 5.08$, $p = 0.042$) by a mean of 0.04 (figure 3*b*). There was no significant effect of obstacle height on landing distance ($F(1, 13) = 3.05$, $p = 0.104$; figure 3*c*), lead apex$_h$ ($F(1, 13) = 0.001$, $p = 0.971$; figure 3*d*) or trail apex$_h$ ($F(1, 13) = 4.52$, $p = 0.053$; figure 3*e*).

As all identified main effects of baseline position (take-off distance, landing distance and lead apex$_h$) represented systematic displacements in the direction of the offset baseline for B− and B+ relative to B0, we conducted a *post hoc* repeated-measures ANOVA for each of these three parameters to test for

**Table 2.** Percentage of obstacle traversals for which the Toe marker was the clearance marker. Results grouped by baseline condition and then by obstacle height.

| factor | lead leg (% trials) | trail leg (% trials) |
|---|---|---|
| baseline condition | | |
| Bx | 34.8 | 100 |
| B+ | 42.9 | 98.2 |
| B0 | 31.3 | 99.1 |
| B− | 20.5 | 100 |
| obstacle height | | |
| low | 35.7 | 99.6 |
| high | 29.0 | 99.1 |

symmetry in the magnitude of the displacement towards the landing side (B+) versus towards the approach side (B−). No significant main or interaction effect was found for baseline condition or obstacle height ($p = 0.13$–$0.84$). Thus, the shifts in foot placement forward for B+ and backward for B− did not differ significantly in magnitude.

## 3.2. Effect of baseline condition on vertical distances

The trail leg clearance marker was the Toe marker in more than 98% of trials for all individual baseline conditions. The lead leg clearance marker was the Heel marker for the majority of trials and was affected by baseline condition, with the percentage of trials in which the Toe marker was the lead crossing marker increasing B− → B0 → B+ from 20.5 to 42.9% as the obstacle baseline moved from the take-off to the landing side of the obstacle (table 2).

Trail leg apex height was always greater than lead leg apex height (figure 4a,b), and trail leg foot clearance was always greater than lead leg foot clearance (figure 4c,d). There were main effects of baseline condition on lead apex$_v$ ($F(3, 39) = 3.48$, $p = 0.025$), trail apex$_v$ ($F(3, 39) = 4.30$, $p = 0.01$) and trail clearance ($F(3, 39) = 3.37$, $p = 0.028$). In all three cases, the value for B+ was significantly greater than the value for the two other conditions (figure 4a,b,d). There was no main effect of condition on lead clearance ($F(3, 39) = 2.39$, $p = 0.083$; figure 4c). No significant differences in the mean or variance of lead apex$_v$, trail apex$_v$, lead clearance or trail clearance were found between Bx and B0.

Lead clearance was not significantly affected by obstacle height ($F(1, 13) = 3.67$, $p = 0.078$; figure 4c). Trail clearance was greater for low obstacles than for high obstacles ($F(1, 13) = 7.43$, $p = 0.017$; figure 4d). Both lead apex$_v$ and trail apex$_v$ were greater for high obstacles than for low obstacles ($F(1, 13) = 133.38$, $p < 0.001$ and $F(1, 13) = 11.81$, $p = 0.004$, respectively; figure 4a,b), but the mean difference between high and low obstacle values was larger for lead apex$_v$ (0.06) than trail apex$_v$ (0.03).

## 3.3. Effect of baseline condition on approach and traversal speed

Approach step speed (Fr $1.03 \pm 0.11$, equivalent to a dimensional speed of $2.95 \pm 0.31$ m s$^{-1}$) was greater than traversal step speed, but neither speed was affected by baseline condition ($F(3, 39) = 0.81$, $p = 0.477$ and $F(3, 39) = 0.90$, $p = 0.448$, respectively; figure 5a,b).

There was no significant effect of obstacle height on approach speed ($F(1, 13) = 0.36$, $p = 0.561$). Traversal speed was greater when traversing low obstacles (Fr $0.85 \pm 0.12$) than when traversing high obstacles (Fr $0.83 \pm 0.11$; $F(1, 13) = 18.12$, $p = 0.001$; figure 5b). As there was no identified effect of obstacle height on step length ($F(1, 13) = 3.20$, $p = 0.097$; figure 3a), the decreased traversal speed when crossing high obstacles would be expected as the vertical distance that must be gained and then lost during the ballistic phase of the traversal is greater.

Speed of travel was not directly controlled during the experiment; participants instead attempted to regulate their own approach speed to match the target speed they had attained running along the track with no obstacle. Although there was no significant difference in approach speed or traversal speed found between obstacle heights or conditions, there was nevertheless inevitably variation in relative approach speed within and between participants. Step length is known to increase with speed of

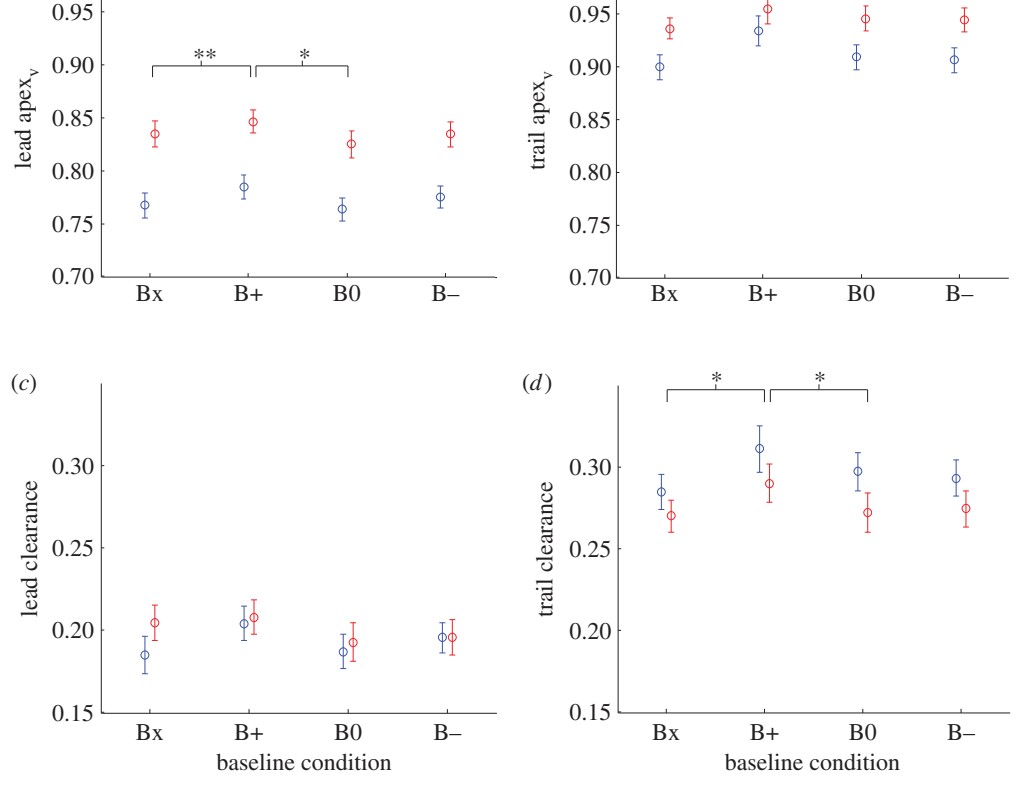

**Figure 4.** Effect of obstacle condition on vertical parameters. Lead (*a*) and trail (*b*) foot trajectory apex vertical height for the four obstacle conditions. Lead (*c*) and trail (*d*) foot clearance for the four obstacle conditions. Legend as in figure 3.

travel [41,42] during running and so some positive correlation might be expected between step length and approach speed. A significant correlation was indeed present ($p \leq 0.001$, $r = 0.58$; figure 5*c*). Assuming a causal (linear regression) model with approach speed as the independent variable, approach speed accounted for 34% of the variation in step length. Similarly, approach speed accounted for 30% of the variation in take-off distance (figure 5*d*).

## 3.4. Summary of baseline condition effects

There were no significant differences between Bx and B0 in the mean value or variability of any measured parameter, indicating that the absence of an obstacle baseline did not affect the kinematics of the traversal. However, altering the baseline position resulted in significant changes to traversal kinematics that could not be accounted for by altered mechanical requirements. These changes were primarily to the horizontal rather than the vertical position of trajectory features. All parameters describing the horizontal position of the limb trajectory, with the exception of trail apex$_h$, were shifted in the direction of the baseline offset. In the vertical direction, trail leg foot clearance and the apex height of both the lead and trail leg Toe markers were slightly but significantly greater for B+ than for two of the three other conditions. Toe apex height and clearance were greater for the trail leg than for the lead leg.

## 4. Discussion

Although placing the baseline directly beneath the obstacle had no effect on any of the measured characteristics of limb trajectory, displacing the baseline altered both horizontal and vertical measures of limb trajectory, indicating that information about baseline position was used for control. These alterations occurred despite the fact that the limb trajectories used when no baseline was present (Bx) could also have been used to traverse the obstacle successfully in all conditions with the baseline present: Not only were take-off and landing distances for all Bx trials at least double the displaced baseline distance from the obstacle, but, in absolute terms, they would have provided a safety margin

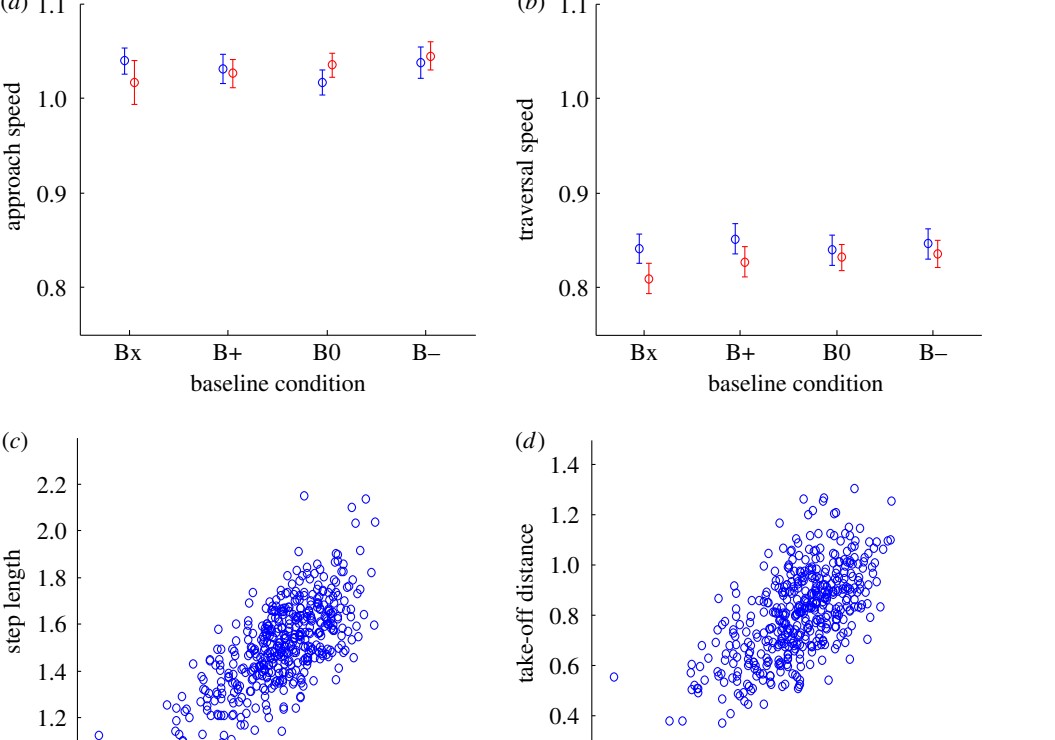

**Figure 5.** Approach speed and its relationship to step length and take-off foot placement. Approach speed (*a*) and traversal speed (*b*) for the four obstacle conditions. Legend as in figure 3. Step length (*c*) and take-off distance (*d*) plotted against approach speed for all trials.

much greater than used by running humans targeting foot placement positions at comparable speeds [3]. Given that an altered trajectory was not required to avoid physical contact with the baseline nor maintain an adequate safety margin, it is difficult to account for the observed changes as serving a useful purpose. Apex heights either remained the same or increased, making it unlikely that the changes minimized the metabolic work done performing the manoeuvre.

Although the magnitude was relatively small, changes to limb trajectories in response to a displaced baseline were made in the direction required to maintain or increase the safety margin. It is possible that they were an artefactual response revealing the nature of a control law that would prevent physical contact should both the baseline displacement and the magnitude of response have been larger. Symmetric displacement of the baseline by the same distance on either side of the obstacle in B− and B+ resulted in the symmetric horizontal displacement of limb trajectories in the same direction as the baseline but by a much smaller distance (0.04–0.06 for a relative baseline displacement of 0.25). Although this suggests a very simple positive relationship might exist between baseline displacement and trajectory displacement, a linear extrapolation of the response, given that the baseline was moved by greater than twice the distance used for the study, would not be sufficient to prevent physical contact. We conclude that a relationship, should it exist, would have to be nonlinear or discontinuous. The vertical measures of the trajectory were either unchanged or increased with baseline displacement. This might be expected, given that the foot follows an arc over the obstacle so shifting the trajectory horizontally would reduce clearance unless apex height was increased. Reduced horizontal take-off distances (as also observed with baseline displacement) have previously been reported to increase the risk of contacting the obstacle with the foot during the traversal step [43], so increased clearance may reduce the likelihood of a destabilizing obstacle contact.

It is possible that the changes to trajectories measured for B− and B+ were caused by an altered perception of the obstacle bar position rather than an explicit characteristic of a control scheme in receipt of accurate information. The relatively small effect size might reflect a judgement based on the low weighting of perceptual information from the baseline combined with a higher weighting of that

from the raised bar: when conflicting information for the guidance of locomotion is provided by different visual cues [44] or by different senses [45–49], the resultant behaviour can be explained by a weighted cue combination. In this study, however, all participants were able to traverse the obstacle easily with only the raised bar present and introducing an identical bar as a baseline directly below the obstacle did not change traversal kinematics. Hence, it seems probable that both bars were sensed accurately and the altered trajectories associated with baseline displacement were not due to sensing error. Although the two bars were clearly behaviourally grouped as a single obstacle, to be traversed within a single leap, it is not known whether they were perceptually grouped as one higher-order obstacle unit or identified as two separate bars. Regardless of perceptual grouping, our results showed that additional visual information was used to control leaping kinematics even though it was not required for obstacle traversal and imposed no new constraints. Similar observations have been made for walking over other types of obstacles in other contexts, but in each case, the additional information provided had been intended to modify the mechanical outcome of obstacle contact [22,23] or had physically intersected with the top boundary of the obstacle in a manner intended or concluded to have altered the accuracy and/or precision of its perceived location [24–31]. Our findings suggest that integration of geometric information can occur without altered sensing of either the top edge or of the mechanical requirements of the traversal.

The significant effect of baseline position on horizontal foot trajectory features identified here differs from the conclusion of Worden *et al.*, who report no effect of baseline position for a similar study conducted in walk [32]. It is possible that the risk of accidental obstacle contact was perceived to be greater in our protocol due to the faster gait and larger obstacles, or that the greater distance between the raised bar and base bar influenced the effect of the baseline offset, but it is unclear how these factors would differentially affect control mechanisms in the manner observed. A more likely explanation for the differing findings is that the horizontal distance by which the baseline can be offset without mechanically interfering with foot placement is considerably reduced when a walking rather than running gait approach is used (figure 3; [32]) so effect magnitude is likely to be smaller. The mean difference between B− and B+ take-off and landing distances in our study was up to one-fifth of the difference between baseline positions in the two conditions: if the same proportional effect had occurred in Worden *et al.*'s study, a power analysis based on the reported descriptive statistics suggests that a considerably larger sample size would have been required for a significant effect of baseline position to have been detected, assuming within-participant correlations equivalent to those observed in our study. The most common foot orientations we observed at the instant of crossing (heel lowest point of lead foot; toe lowest point of trail foot) are consistent with the behaviour observed when stepping over raised obstacles in walk [8,50]. The speed control method we adopted, asking participants to replicate a target approach speed, resulted in considerable approach speed variability explaining about one-third of the linear variation in step length and take-off distance (figure 5*c,d*). There were, however, no main effects of the condition, obstacle height or trial number on approach speed (figure 5*a*), so the main results were not confounded by speed variation.

We did not attempt to measure the explicit perception of obstacle position or height. As traversing an obstacle is an action task, the focus of the current study was the resultant locomotor behaviour rather than explicit perceptual judgements. While a corresponding change in perceived obstacle height has been found alongside motor adaptations to increase foot clearance during obstacle traversal [24,25], other studies have found a dissociation between motor responses and explicit perceptual judgement in tasks for which an object's explicitly perceived size is illusorily modified [51–55]. Different experimental methodologies may have contributed to the differing findings, as both continuous online feedback during the task and repeated task exposure appear to facilitate motor adaptation without associated perceptual adaptation, and hence differentiation between the two metrics [24,54]. The effect of baseline position on the explicit perception of obstacle height and egocentric distance, as well as the conditions under which perceptual grouping of multiple obstacle elements occurs and whether the presence of such grouping influences traversal behaviour, remain key potential directions for future investigation.

# 5. Conclusion

Our findings suggest that visuomotor integration of geometric information to modulate limb kinematics when leaping over raised obstacles can occur without altered sensing of either the obstacle top edge or of the mechanical constraints on limb trajectories for the traversal. The control laws relating obstacle characteristics to traversal behaviours hence do not appear to be fully explained by perceived

mechanical requirements and consequences of obstacle contact, instead implying a more fundamental principle of control.

Ethics. Ethical approval for the protocol was provided by the University of Bristol faculty ethics committee. Research was conducted according to the principles expressed in the Declaration of Helsinki and informed written consent was obtained from all participants.

Data accessibility. The datasets supporting this article have been uploaded as electronic supplementary material.

Authors' contributions. K.A.J.D. and J.F.B. designed the study, interpreted the data, wrote the manuscript and gave final approval for publication. K.A.J.D. collected and analysed the data.

Competing interests. The authors declare no competing interests.

Funding. This research was supported by an Engineering and Physical Sciences Research Council DTG (grant no. EP/P50483X/1) to K.A.J.D. and by a Wellcome Trust equipment grant to Bristol Vision Institute.

Acknowledgements. The authors acknowledge Casimir Ludwig for helpful comments on study design and on an early draft of the manuscript.

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
