## [Peer Review File · Royal Society Open Science]

Review History

RSOS-201877.R0 (Original submission)

Review form: Reviewer 1

Is the manuscript scientifically sound in its present form?

Yes

Are the interpretations and conclusions justified by the results?

Yes

Is the language acceptable?

Yes

Do you have any ethical concerns with this paper?

No

Have you any concerns about statistical analyses in this paper?

No

Recommendation?

Accept with minor revision (please list in comments)

Comments to the Author(s)

Thank you for the opportunity to review this interesting research study. The authors have presented a very well written manuscript that was a pleasure to review.

The primary aim of the study was to explore the influence of spatially remote objects on obstacle clearance during running. This is an interesting experimental paradigm and provides information that will be relevant to many fields. In general, I believe this paper is very well prepared, thoroughly considered and appropriate for publication.

I only have minor questions and these are listed below;

1. Please provide more detail, or a source reference for your approach to determine stance phase
2. Statistics - Please specify the approach to account for multiple comparisons in the analysis

Review form: Reviewer 2

Is the manuscript scientifically sound in its present form?

Yes

Are the interpretations and conclusions justified by the results?

Yes

Is the language acceptable?

Yes

Do you have any ethical concerns with this paper?

No

Have you any concerns about statistical analyses in this paper?

No

Recommendation?

Accept with minor revision (please list in comments)

Comments to the Author(s)

Overall this is a well written, concise article describing an experiment where the presence and location of a baseline under a jump is manipulated to study the effects on an individual's kinematics. I have few changes to request.

Abstract: Inclusion of some numerical results here would be helpful for the reader.

Introduction: Is the stated objective rather an aim?

Methods: This is unusual in its high number of female participants, do you think there may be any sex-related differences present?

The difference in height is quite small - can you comment on how results may potentially change with a greater change in height?

In your supplementary material, the conditions are listed as C1-4, some additional explanation in the SM to reiterate how these relate to the condition names in this section would be helpful.

Results: in this section, some parts are more discussion (e.g. P7 line 47 on). It would be helpful to include some key numerical results in this section, currently these are presented in the text as solely F and p values. Although referring to the figures, it would be clearer for the reader for some key numbers to be in this section.

Discussion: P8 line 54 on: It would be clearer to restate how the results support or refute the stated hypotheses/address the study aim/objectives here in the first instance, it is a little unclear what the headline findings are from this initial paragraph.

Decision letter (RSOS-201877.R0)

Dear Dr Daniels

On behalf of the Editors, we are pleased to inform you that your Manuscript RSOS-201877 "Visuomotor control of leaping over a raised obstacle is sensitive to small baseline displacements" has been accepted for publication in Royal Society Open Science subject to minor revision in accordance with the referees' reports. Please find the referees' comments along with any feedback from the Editors below my signature.

Please submit your revised manuscript and required files (see below) no later than 7 days from today's (ie 27-Jan-2021) date. Note: the ScholarOne system will 'lock' if submission of the revision is attempted 7 or more days after the deadline. If you do not think you will be able to meet this deadline please contact the editorial office immediately.

Best regards,

on behalf of the Associate Editor, and Professor Kevin Padian (Subject Editor)
openscience@royalsociety.org

Editor Comments to Author:

Thank you for your submission. As you see, the reviewers are largely happy with your manuscript and have offered a few minor things to address. We're pleased to accept it pending your revisions. Best wishes.

Reviewer comments to Author:

Reviewer: 1

Comments to the Author(s)

Thank you for the opportunity to review this interesting research study. The authors have presented a very well written manuscript that was a pleasure to review.

The primary aim of the study was to explore the influence of spatially remote objects on obstacle clearance during running. This is an interesting experimental paradigm and provides information that will be relevant to many fields. In general, I believe this paper is very well prepared, thoroughly considered and appropriate for publication.

I only have minor questions and these are listed below;

1. Please provide more detail, or a source reference for your approach to determine stance phase
2. Statistics - Please specify the approach to account for multiple comparisons in the analysis

Reviewer: 2

Comments to the Author(s)

Overall this is a well written, concise article describing an experiment where the presence and location of a baseline under a jump is manipulated to study the effects on an individual's kinematics. I have few changes to request.

Abstract: Inclusion of some numerical results here would be helpful for the reader.

Introduction: Is the stated objective rather an aim?

Methods: This is unusual in its high number of female participants, do you think there may be any sex-related differences present?

The difference in height is quite small - can you comment on how results may potentially change with a greater change in height?

In your supplementary material, the conditions are listed as C1-4, some additional explanation in the SM to reiterate how these relate to the condition names in this section would be helpful.

Results: in this section, some parts are more discussion (e.g. P7 line 47 on). It would be helpful to include some key numerical results in this section, currently these are presented in the text as solely F and p values. Although referring to the figures, it would be clearer for the reader for some key numbers to be in this section.

Discussion: P8 line 54 on: It would be clearer to restate how the results support or refute the stated hypotheses/address the study aim/objectives here in the first instance, it is a little unclear what the headline findings are from this initial paragraph.

===PREPARING YOUR MANUSCRIPT===

===PREPARING YOUR REVISION IN SCHOLARONE===

- An individual file of each figure (EPS or print-quality PDF preferred [either format should be produced directly from original creation package], or original software format).
 - An editable file of each table (.doc, .docx, .xls, .xlsx, or .csv).
 - An editable file of all figure and table captions.
- Note: you may upload the figure, table, and caption files in a single Zip folder.
- Any electronic supplementary material (ESM).
 - If you are requesting a discretionary waiver for the article processing charge, the waiver form must be included at this step.
 - If you are providing image files for potential cover images, please upload these at this step, and inform the editorial office you have done so. You must hold the copyright to any image provided.
 - A copy of your point-by-point response to referees and Editors. This will expedite the preparation of your proof.

- Ensure that your data access statement meets the requirements at <https://royalsociety.org/journals/authors/author-guidelines/#data>. You should ensure that you cite the dataset in your reference list. If you have deposited data etc in the Dryad repository, please only include the 'For publication' link at this stage. You should remove the 'For review' link.
- If you are requesting an article processing charge waiver, you must select the relevant waiver option (if requesting a discretionary waiver, the form should have been uploaded at Step 3 'File upload' above).
- If you have uploaded ESM files, please ensure you follow the guidance at <https://royalsociety.org/journals/authors/author-guidelines/#supplementary-material> to include a suitable title and informative caption. An example of appropriate titling and captioning may be found at https://figshare.com/articles/Table_S2_from_Is_there_a_trade-off_between_peak_performance_and_performance_breadth_across_temperatures_for_aerobic_scope_in_teleost_fishes_/3843624.

Author's Response to Decision Letter for (RSOS-201877.R0)

See Appendix A.

Decision letter (RSOS-201877.R1)

Dear Dr Daniels,

It is a pleasure to accept your manuscript entitled "Visuomotor control of leaping over a raised obstacle is sensitive to small baseline displacements" in its current form for publication in Royal Society Open Science.

You can expect to receive a proof of your article in the near future. Please contact the editorial office (openscience@royalsociety.org) and the production office (openscience_proofs@royalsociety.org) to let us know if you are likely to be away from e-mail contact – if you are going to be away, please nominate a co-author (if available) to manage the proofing process, and ensure they are copied into your email to the journal.

on behalf of the Associate Editor and Professor Kevin Padian (Subject Editor)
openscience@royalsociety.org

Appendix A

Response to reviewers

We thank both reviewers for their positive comments and valuable feedback on our study. We have responded to each of their points individually in green below and include a reference list at the end of this document. Red text has been used in the marked-up version of the revised manuscript to highlight alterations to the original text.

Reviewer 1

Thank you for the opportunity to review this interesting research study. The authors have presented a very well written manuscript that was a pleasure to review.

The primary aim of the study was to explore the influence of spatially remote objects on obstacle clearance during running. This is an interesting experimental paradigm and provides information that will be relevant to many fields. In general, I believe this paper is very well prepared, thoroughly considered and appropriate for publication.

1. Please provide more detail, or a source reference for your approach to determine stance phase. The start of stance phase was defined as the first frame in which the horizontal velocity of the toe marker decreased below 0.006 m/s and the vertical height of the toe marker was below 0.05 m; the end of stance phase as the last frame before the horizontal velocity of the toe marker next increased above 0.006 m/s and the vertical height of the toe marker next increased above 0.05 m. This information is now included in the second paragraph of the 'Calculation of kinematic parameters' Methods subsection:

Stance phases were identified automatically using an algorithm incorporating vertical height and horizontal velocity of a marker placed on the fifth metatarsal head of the relevant foot (horizontal velocity of marker <0.006 m/s and vertical height <0.05 m).

These thresholds were selected by adjusting their values until the identified touchdown and toe-off frames of both normal running gait and the obstacle traversal steps were within 3 frames (± 0.025 s) of the visually identified first and last frames in which the foot was in contact with the ground. As spatial rather than temporal variables were extracted from stance phase (with the exception of traversal speed), our study's results are not sensitive to the precise timing of touchdown and toe-off identification.

2. Statistics - Please specify the approach to account for multiple comparisons in the analysis. The family-wise error rate is controlled for within the *post hoc* pairwise comparisons that were conducted where we identified an ANOVA main effect ('Statistical analysis' subsection, paragraph 2), but we did not control for the fact that we performed multiple ANOVAs (11 in total) as these analyses were planned and complementary. We now specify this explicitly in the manuscript text – the relevant section now reads:

Greenhouse-Geisser's adjustment to the degrees of freedom was used when Mauchley's test of sphericity was significant, and Tukey's HSD post hoc tests were used to identify pairwise differences

*between groups when main effects were found. Bartlett's test for equality of variances was used to test for equal variance of the dependent variable between condition Bx and B0 for each obstacle height. Significance was accepted at $\alpha = .05$ and the symbols *, ** and *** were used in figures to represent $p < .05$, $p < .01$ and $p < .001$ respectively. No family-wise error rate correction was performed for ANOVA main effects because these analyses were planned and complementary.*

We note that the decision not to control for multiple ANOVAs did not affect the conclusions of the study: even employing the Bonferroni correction for multiple comparisons, which is excessively conservative for complementary tests, only one main effect of obstacle condition (take-off distance) would no longer be identified as significant at a family-wise alpha level of 0.05.

Reviewer 2

Overall this is a well written, concise article describing an experiment where the presence and location of a baseline under a jump is manipulated to study the effects on an individual's kinematics. I have few changes to request.

Abstract: Inclusion of some numerical results here would be helpful for the reader.

As the abstract cannot be extended beyond 200 words, we are not able to add additional material here without removing text that we feel is essential for the reader to understand the study, particularly as there is no single numerical finding that fully encapsulates our conclusions. Instead, we now include appropriate numerical results in the Results section of the manuscript, as per the reviewer's later suggestion, and detail these additions below in response to their second-last comment.

Introduction: Is the stated objective rather an aim?

We have changed the relevant word from 'objective' to 'aim'. The sentence now reads:

The aim of this study was to investigate whether a simple geometric manipulation, spatially remote from the geometric feature signalling height and not affecting the perception of risk, would still elicit a change in limb kinematics during a running leap manoeuvre.

Methods: This is unusual in its high number of female participants, do you think there may be any sex-related differences present?

We are not aware of any evidence to suggest that sex differences are present in visuomotor control of locomotion over obstacles. The primary sex-related factor that would be relevant to our study is that of leg length, and we control for this in the experimental design by scaling all obstacle heights and spatial parameters to the participant's greater trochanter height. It is possible that group differences in athletic capacity might influence traversal kinematics when leaping over higher obstacles, but those traversed within this study were comfortably within the capabilities of all participants of both sexes.

The difference in height is quite small - can you comment on how results may potentially change with a greater change in height?

We interpret this comment to be in reference to the height of the raised bar traversed by the participants.

The primary purpose of presenting two bar height conditions was not to investigate the effect of obstacle height on the measured parameters but simply to minimise the likelihood of the participants learning the height of the obstacle with repeated trials. There is a considerable existing literature on the effect of obstacle height on traversal kinematics, and the effects we observed on foot-obstacle distances, foot apex heights and foot clearances have been previously noted in earlier studies of obstacle traversal from walking gait (1–4). Whilst we might expect to see greater divergence in these parameters between conditions if we increased the height difference, we have no reason to believe that the observed effects of baseline position would be altered. It is conceivable that the physical proximity of the raised bar to the baseline might modulate the strength of the baseline offset effect, or that reduced safety margins at the upper height limit of participant leap capacity might modify control strategies, but this is speculation. We note the potential influence of obstacle size on our results in the fourth paragraph of the Discussion section - the relevant sentence now reads:

It is possible that the risk of accidental obstacle contact was perceived to be greater in our protocol due to the faster gait and larger obstacles, or that the greater distance between the raised bar and base bar influenced the effect of the baseline offset, but it is unclear how these factors would differentially affect control mechanisms in the manner observed.

In your supplementary material, the conditions are listed as C1-4, some additional explanation in the SM to reiterate how these relate to the condition names in this section would be helpful.

We now include a 'ReadMe' tab in the Supplementary Data spreadsheet which describes the coding of the obstacle height and condition factors with reference to the terminology used in the main manuscript.

Results: in this section, some parts are more discussion (e.g. P7 line 47 on).

We have edited this sentence and moved it to the Discussion. The new sentence (Discussion section, paragraph 4) reads:

The most common foot orientations we observed at the instant of crossing (heel lowest point of lead foot; toe lowest point of trail foot) are consistent with the behaviour observed when stepping over raised obstacles in walk (8,41).

It would be helpful to include some key numerical results in this section, currently these are presented in the text as solely F and p values. Although referring to the figures, it would be clearer for the reader for some key numbers to be in this section.

We report primarily F and p values in the manuscript text because there were two factors with main effects included in the ANOVA models, so eight individual means and standard deviations would need to be reported for each test and we consider these to be better presented within figures. We now include numerical results for clearance marker overall percentages, mean effect of obstacle height on take-off distance, approach step speed (including dimensional equivalent) and traversal step speed.

Discussion: P8 line 54 on: It would be clearer to restate how the results support or refute the stated hypotheses/address the study aim/objectives here in the first instance, it is a little unclear what the headline findings are from this initial paragraph.

We have revised and restructured the initial paragraph so it now leads with the 'headline' result that addresses the main study aim. The paragraph now reads:

Although placing the baseline directly beneath the obstacle had no effect on any of the measured characteristics of limb trajectory, displacing the baseline altered both horizontal and vertical measures of limb trajectory indicating that information about baseline position was used for control. These alterations occurred despite the fact that the limb trajectories used when no baseline was present (Bx) could also have been used to traverse the obstacle successfully in all conditions with the baseline present: Not only were take-off and landing distances for all Bx trials at least double the displaced baseline distance from the obstacle, but in absolute terms they would have provided a safety margin much greater than used by running humans targeting foot placement positions at comparable speeds (3). Given that an altered trajectory was not required to avoid physical contact with the baseline nor maintain an adequate safety margin it is difficult to account for the observed changes as serving a useful purpose. Apex heights either remained the same or increased, making it unlikely that the changes minimised the metabolic work done performing the manoeuvre.

References

1. Austin GP, Garrett GE, Bohannon RW. Kinematic analysis of obstacle clearance during locomotion. *Gait Posture* [Internet]. 1999 Oct;10(2):109–20. Available from: <http://www.ncbi.nlm.nih.gov/pubmed/10502644>
2. Sparrow WA, Shinkfield AJ, Chow S, Begg RK. Characteristics of gait in stepping over obstacles. *Hum Mov Sci* [Internet]. 1996 Aug;15(4):605–22. Available from: <http://linkinghub.elsevier.com/retrieve/pii/016794579600022X>
3. Patla AE, Rietdyk S. Visual control of limb trajectory over obstacles during locomotion: effect of obstacle height and width. *Gait Posture*. 1993;1(1):45–60.
4. Patla AE. Understanding the roles of vision in the control of human locomotion. *Gait Posture* [Internet]. 1997 Feb;5(1):54–69. Available from: <http://linkinghub.elsevier.com/retrieve/pii/S0966636296011095>